# YOLOv7-RAR for Urban Vehicle Detection

**DOI:** 10.3390/s23041801

**Published:** 2023-02-06

**Authors:** Yuan Zhang, Youpeng Sun, Zheng Wang, Ying Jiang

**Affiliations:** School of Mechanical and Electronic Engineering, Nanjing Forestry University, Nanjing 210037, China

**Keywords:** vehicle detection, ACmix, RFLA, YOLOv7

## Abstract

Aiming at the problems of high missed detection rates of the YOLOv7 algorithm for vehicle detection on urban roads, weak perception of small targets in perspective, and insufficient feature extraction, the YOLOv7-RAR recognition algorithm is proposed. The algorithm is improved from the following three directions based on YOLOv7. Firstly, in view of the insufficient nonlinear feature fusion of the original backbone network, the Res3Unit structure is used to reconstruct the backbone network of YOLOv7 to improve the ability of the network model architecture to obtain more nonlinear features. Secondly, in view of the problem that there are many interference backgrounds in urban roads and that the original network is weak in positioning targets such as vehicles, a plug-and-play hybrid attention mechanism module, ACmix, is added after the SPPCSPC layer of the backbone network to enhance the network’s attention to vehicles and reduce the interference of other targets. Finally, aiming at the problem that the receptive field of the original network Narrows, with the deepening of the network model, leads to a high miss rate of small targets, the Gaussian receptive field scheme used in the RFLA (Gaussian-receptive-field-based label assignment) module is used at the connection between the feature fusion area and the detection head to improve the receptive field of the network model for small objects in the image. Combining the three improvement measures, the first letter of the name of each improvement measure is selected, and the improved algorithm is named the YOLOv7-RAR algorithm. Experiments show that on urban roads with crowded vehicles and different weather patterns, the average detection accuracy of the YOLOv7-RAR algorithm reaches 95.1%, which is 2.4% higher than that of the original algorithm; the AP50:90 performance is 12.6% higher than that of the original algorithm. The running speed of the YOLOv7-RAR algorithm reaches 96 FPS, which meets the real-time requirements of vehicle detection; hence, the algorithm can be better applied to vehicle detection.

## 1. Introduction

Traffic congestion is a common occurrence in cities. On one hand, it is related to urban road design; on the other hand, it is related to artificial driving. Drivers on the road completely depend on their driving experience and can be driving for a long time, which can cause the visual fatigue of drivers and car accidents. It is of great significance to study a method that assists or even replaces the human eye and completes the vehicle’s automatic recognition and detection reliably.

Vehicle detection and recognition is a popular research direction in computer vision, which has a wide application prospect in automatic driving. However, the real-time acquisition of road vehicle pictures by onboard cameras in the process of image acquisition is affected by camera angles and the distance between bodies; there will be problems such as block, blur, dark light, and the small size of the target object. Thus, the recognition rate is low. In order to improve the recognition rate, Zha et al. [1] studied the image information of vehicles in parking lots, trained the classifier by manually extracting features, and matched the interested vehicle features to obtain better recognition results. Amit et al. [2] proposed a strong classifier based on a machine learning algorithm that uses more features to form a better decision boundary and fewer features to exclude a large number of negative samples and trains a discriminative weak classifier with Haar features a generative weak classifier with HOG features. Taking the AdaBoost algorithm as a bridge, the recognition rate reached 95.7%. However, the above algorithms are based on the application of machine learning algorithms, requiring a high level of manual processing, such as manual feature extraction and classifier design, and the maximum processing speed of the algorithm is not more than 40 FPS, which is low and cannot be applied to urban roads with faster driving speeds.

With the development of deep learning, the extraction method of image features has changed. Different from traditional manual feature extraction, the method based on deep learning can independently extract features and learn [3,4]. Based on the idea of deep learning, Cao Shiyu et al. [5] used a selective search algorithm to obtain the candidate regions of the sample as the input of the network training network. The convolutional features were successfully used to replace the traditional manual features and achieved good results in view of the problem that traditional vehicle target detection needs to manually select appropriate features for different scenes. I.O.D. Oliveira et al. [6,7] used a double-flow convolutional neural network to train and learn the high-resolution dataset Vehicle-Rear, which solved the problem of non-overlapping cameras identifying vehicles. All the above studies show that feature extraction by convolutional algorithms completely avoids traditional manual feature extraction. The development of object detection algorithms in deep learning has gradually been divided into two categories: One is a two-stage detection algorithm based on the selective search method to select the region proposal. The basic idea of this kind of algorithm is based on the R-CNN algorithm [8,9,10]. Firstly, selective search is used to select about 2000 region proposals per image. Since the network structure can only accept proposals of the same size, each proposal needs to be warped to 227 × 227. For each proposal, CNN is used to extract features, and then these features are used to train an SVM classifier to obtain the corresponding category score [11,12,13,14,15]. NMS (non-maximum suppression) is used to remove some redundant candidate boxes, and bounding box regression is used to fine-tune the candidate boxes. Then, the detection task of the two-stage algorithm is complete. The two-stage target detection algorithm is mainly characterized by high detection accuracy, but the running speed of the algorithm rarely exceeds 30 FPS, and it needs to consume a lot of storage space [16,17,18,19,20,21]. The representative algorithms of two-stage algorithms are R-CNN, Fast R-CNN, Faster R-CNN, and Mask R-CNN [22,23,24,25]. Compared with the advanced performance of the object detection algorithm at that time in general object detection and general performance in vehicle detection, Quanfu Fan et al. [15] revealed a vehicle detection technique using Faster R-CNN at the 2016 IEEE Intelligent Vehicles Symposium. It gave a research direction for vehicle detection. Li Wang et al. [26] made an improvement on Faster R-CNN, which improved the recognition accuracy by 9.5% compared with the original network, and the detection speed reached 9–13 FPS. Another algorithm is based on end-to-end detection. The core idea of this algorithm is to transform the problem of object classification into the regression problem of the object detection box and bounding box. The algorithm does not need to select the candidate box, and the result can be obtained by direct detection. This idea omits the complex candidate box selection and calculation steps in the two-stage algorithm. Therefore, the running speed of the algorithm is far faster than that of the two-stage detection algorithm [27,28,29]. The representative algorithms are the YOLO algorithm and the SSD algorithm. In the two-stage algorithm, it is difficult to improve the speed of vehicle detection, so researchers have turned their attention to single-stage detection algorithms. Jun Sang et al. [30] studied and created the improved YOLOv2. Through the K-means ++ algorithm, the appropriate anchor box was selected for the collected dataset, and the YOLOv2_V algorithm was proposed. The accuracy of the algorithm reached 94.78%. Fukai Zhang et al. [31] proposed DP-SSD by improving the SSD algorithm, and the accuracy of the algorithm reached 75.43% at the running speed of 50.47 FPS. In 2016, Joseph Redmon of the University of Washington proposed the YOLOv1 algorithm [32], and the SSD algorithm was proposed by Wei Liu at the ECCA in 2016 [33,34]. Its basic framework improves on the YOLO algorithm and draws on the anchor mechanism in Faster-RCNN. A prior box is generated on the feature map for prediction, and similar to YOLOv3, an anchor is generated on the feature map at multiple different scales [35]. At present, the YOLO algorithm is widely used in industrial production and has become the mainstream target detection algorithm. Therefore, the application algorithm framework used in this paper is the YOLO algorithm. At present, the performance of this algorithm is the best in YOLOv7, as proposed by Wang et al. in 2022 [36]. However, there is still room for improvement in the detection accuracy of the algorithm.

This paper proposes three directions of improvement mechanisms. First of all, it is to improve the network model architecture. In most literature on high-speed architecture design, the model’s parameter number, computation amount, and computation density are mainly considered. Ma et al. [37] also analyzed the influence of the input–output channel ratio, the number of architecture branches, and operation by element on the network reasoning speed based on the characteristics of memory access cost. Dollar P et al. [38] gave extra consideration to activation when scaling the model, that is, more consideration to the number of elements in the output tensor of the convolution layer. The above gradient analysis methods enable faster and more accurate reasoning, and the design of the ELAN network leads to the conclusion that a deeper network can learn and converge effectively by controlling the shortest and longest gradient paths. Based on this conclusion, the Res3Unit module is proposed in this paper.

Second, the addition of attention mechanisms. Traditional attention mechanisms, such as CBAM and SENet, are usually used in the enhancement of convolutional modules [39,40]. Recently, self-attention modules have been proposed to replace traditional convolutions such as SAN and BoTNet. However, the relationship between self-attention mechanisms and convolution has not been discovered and utilized. Xuran Pan et al. [41] found that the two modules are heavily dependent on the same 1 × 1 convolution operation, so they proposed an ACmix module that can reuse and share the features obtained by the two modules and aggregate the intermediate features. Compared with pure convolution or self-attention modules, this module has less overhead. Hence, this article adds ACmix to the network structure.

Finally, in order to enhance the receptive field of the network for distant small targets, the RFLA_Gauss_ module is introduced. Enhancing the receptive field of the network to the target object is the RFBNet model proposed by Songtao Liu et al. [42], first published in ECCV 2018. This model mainly adds a dilated convolutional layer on the basis of inception to effectively increase the receptive field. In this paper, we introduce a new module, RFLA_Gauss_, which was proposed by Chang Xu et al. [43] in 2022, aiming at the characteristics of small objects in perspective, such as fewer pixels in the whole image and limited features that can be collected. The real box does not overlap with almost all anchor boxes (that is, IoU = 0) and does not contain any anchor points, resulting in the lack of positive samples of small objects. New prior knowledge based on Gaussian distribution was introduced, and a label assignment strategy based on the Gaussian receptive field (RFLA) was established, which solved the problem of small object recognition. Therefore, this paper will introduce the model structure to improve the detection effect of the algorithm model for small targets.

In this paper, the YOLOv7 algorithm is improved and the YOLOv7-RAR algorithm is proposed. To solve the problem of insufficient fusion of the nonlinear characteristics of the network, the Res3Unit structure is proposed to reconstruct the backbone network of YOLOv7 and to solve the problem that the network is weak in vehicle target positioning. We add the plug-and-play module ACmix between the backbone network and the detection head. The RFLA_Gauss_ module is used to solve the problem that the receptive field of the original network shrinks with the deepening of the network model, and four groups of ablation experiments are conducted to compare the performance of the improved model.

## 2. Problem Description

At present, the vehicle detection and recognition algorithm is more inclined to high-performance algorithms; that is, the algorithm has the characteristics of high accuracy and high-speed image processing. Among the current excellent object detection algorithms, both single-stage detection algorithms and two-stage detection algorithms have mediocre performance in vehicle detection and recognition. Specifically, the algorithms have poor performance in detection speed, which does not meet the needs of the high real-time performance of vehicles. Table 1 shows the performance of some algorithms using the UA-DETRAC dataset; as can be seen from the table, the real-time performance of most algorithms is not high, and the detection accuracy of algorithms with high real-time performance is low. However, the YOLO algorithm gradually improves the algorithm performance with model iteration. At present, the YOLO algorithm has been updated to YOLOv7, and the excellent architecture of YOLOv7 is used for further improvement in order to obtain better algorithm performance in the detection vehicle.

## 3. Methodology

### 3.1. YOLOv7 Algorithm

The authors of YOLOv7 are Chien-Yao Wang, Alexey Bochkovskiy, and Hong-yuan Mark Liao. One of the improvements of YOLOv7 is that the activation function is changed from Leakrelu to Swish. Other basic modules are optimized by using the residual design idea for reference, but the basic architecture of the network has not changed much and still includes three parts: backbone, neck, and head.

#### 3.1.1. Backbone

DarkNet, the basic backbone network of the YOLO algorithm, was built by Joseph Redmon. Other versions of the YOLO algorithm are optimized on its architecture. The backbone network of YOLOv7 includes the CBS, E-ELAN, MP, and SPPCSPC modules. CBS, as the most basic module, is integrated into other modules.

#### 3.1.2. Feature Fusion Zone

The feature fusion layer of the network is to enable the network to better learn the features extracted from the backbone network. The features of different granularities are learned separately and merged in a centralized way so as to learn as many image features as possible.

#### 3.1.3. Detection Head

The YOLOv7 algorithm follows the advantages of previous algorithms and retains three detection heads, which are used to detect and output the predicted category probability, confidence, and predicted frame coordinates of the target object. The detection heads output three feature scales: 20 × 20, 40 × 40, and 80 × 80. The target scales detected by the three scales respectively correspond to a large target, a medium target, and a small target.

### 3.2. YOLOv7-RAR Algorithm

#### 3.2.1. Backbone

The original backbone network is first stacked by four CBS; four convolution operations are performed on the input image to extract the underlying features, and then the fine-grained features are extracted by the MP and E-ELAN modules. However, such a structure will still use a lot of repeated feature information and lose more fine-grained features [44,45]. It is not good for the network to learn more nonlinear features. In order to further reduce the use of repeated features and deepen the extraction of fine-grained feature information, this paper proposes an improved network module, Res3Unit, based on ELAN. Its main idea is to let the network obtain as many nonlinear features as possible and reduce the use of repeated features. The network module sets the structure of multiple fusion branches, which will reduce the use of repeated features and fuse the features collected by the upper layer to be more fine-grained [36]. The Res3Unit structure is shown in Figure 1.

A picture of a car was selected for testing, and the results are shown in Figure 2. Three stage features that need to be sampled are selected, namely, stage6_E-ELAN_features, stage8_E-ELAN_features, and stage12_SPPCSPC_features. Compared with the original backbone network to extract the nonlinear features of the image, it is found that the improved backbone network can extract the nonlinear features of the vehicles in the image more fully and clearly, indicating the effectiveness of the improved algorithm.

#### 3.2.2. Mixed Attention Mechanism

The self-attention module uses the weighted average operation based on the context of the input features to dynamically calculate the attention weight through the similarity function between the relevant pixel pairs. This flexibility allows the attention module to adaptively focus on different areas and capture more features. The study of early attention mechanisms such as SENet and CBAM shows that self-attention can be used as an enhancement of the convolution module. By decomposing the operations of these two modules, it shows that they largely depend on the same convolution operation. Based on this observation, Xuran Pan et al. proposed a hybrid attention mechanism ACmix module in CVPR 2022. First, a rich set of intermediate features is obtained by mapping the input features using convolution. The intermediate features are then reused and aggregated in different modes (self-attention and convolution, respectively). In this way, ACmix enjoys the advantages of two modules while effectively avoiding two expensive projection operations. 

As shown in Figure 3, the ACmix module is added after the SPPCSPC module of the backbone network to enhance the feature perception and location information of the backbone network on small targets of distant vehicles and reduce the attention to the interference background. As shown in Figure 4 [41], since both modules share the same 1 × 1 convolution operation, only one projection can be performed, and these intermediate feature maps are used for different aggregation operations, respectively.

#### 3.2.3. Enhancing the Network Receptive Field

The RFB module, proposed by Songtao Liu et al., was the earliest one to use the RF module to enhance the receptive field. This module simulates the receptive field of human vision to enhance the feature extraction ability of the network. It uses different convolution kernels and different step sizes to combine different receptive fields, connects 1 × 1 convolutions to reduce the dimensionality, and finally forms a hybrid superposition of different receptive fields. Its module structure is shown in Figure 5 [42].

However, according to the analysis of Guo et al., the feature receptive fields learned at different scales are different, and the superposition and fusion of different receptive fields in the feature fusion area (neck) of YOLOv7 will weaken the multi-scale feature expression, resulting in poor detection effects on smaller targets. For the performance of a small target on the image, when the receptive field of its feature points is remapped back to the input image, the effective receptive field is actually Gaussian-distributed. The gap between the prior uniform distribution and the Gaussian-distributed receptive field will result in a mismatch between the ground truth and the receptive fields of the feature points assigned to it. RFLA, published by Xu et al. in ECCV 2022, effectively solves the receptive field problem of small target recognition. Therefore, the RFLA_Gauss_ module optimization network based on the Gaussian receptive field is introduced in this paper. The principle of the model is shown in Figure 6. Firstly, feature extraction is performed, and then convolution is performed with the Gaussian kernel function. After that, the extracted features are integrated into a feature point, and the Gaussian effective receptive field is obtained.

## 4. Materials and Experiments

### 4.1. Dataset Selection

The dataset used in this paper is UA-DETRAC. UA-DETRAC is a large-scale dataset for vehicle detection and tracking. The UA-DETRAC dataset contains 140,000 images (60% for training and 40% for testing). The image scale is 960 × 540. The dataset is mainly taken from road overpasses in Beijing and Tianjin (Beijing–Tianjin–Hebei scene), and 8250 vehicles and 1.21 million target objects are manually labeled. Figure 7 shows the types of vehicles in the dataset, including cars, buses, vans, and other types of vehicles. Figure 8 shows the distribution and vehicle types of the dataset. The weather conditions are divided into four categories, namely, cloudy, night, sunny, and rainy. Figure 9 shows the initial training samples of part of the training dataset.

### 4.2. Experimental Analysis and Comparison

#### 4.2.1. Environmental Preparation

In this paper, the Pytorch framework is used as the experimental environment for algorithm training. The environment is CUDA v11.2, the Pytorch version is v1.10, the GPU version is NVIDIA GeForce RTX3090, the video memory is 25.4 GB, and the Python version is 3.9. The batch size of each batch of training is set to 32 for a total of 200 training rounds. In order to verify the effectiveness of the three improved methods proposed in this paper, this paper uses four groups of ablation experiments. In the first group of experiments, we only retained the Res3Unit module in the original network architecture and named the module YOLOv7-Res. In the second group of experiments, we only retained ACmix and named the improved model YOLOv7-AC. In the third group of experiments, we only retained the RFLA module and named the model YOLOv7-RF. In the fourth group of experiments, we retained all the improved modules to test the performance of the YOLOv7-RAR model. When training the network, the input image was resized to a uniform size of 640 × 640, the initial learning rate was set to 0.01, and the One Cycle Policy was used to adjust the learning rate. The parameter settings are shown in Table 2. In the experiment, the models are evaluated by the average precision (AP), the AP50 value of IOU greater than 0.5, AP50:90, the number of images detected per second (FPS), and the amount of model calculation (GFLOPs).

#### 4.2.2. Backbone Network Improvement

Comparing the YOLOv7-RAR algorithm backbone network using Res3Unit with the original YOLOv7 backbone network is shown in Figure 10, the obvious improvement is in the model recall; the experimental results are shown in Table 3. After using the Res3Unit module, the average accuracy of AP50 increases by 1.7%, and the value of AP50:90 increases by 1.8%, indicating that the Res3Unit module has a greater improvement on the backbone network performance of the algorithm.

#### 4.2.3. Adding the Mixed Attention Mechanism

The mixed attention mechanism ACmix module is added to the MP module and the E-ELAN module of the original network, and the experimental results of the comparison of the original network are shown in Figure 11 and Table 4. Compared with the original network model performance, the optimization strategy in this paper reduced the model calculation by 10.8%, and the model’s FPS value is also larger, which fully meets the real-time requirements of urban traffic detection. It shows that the performance of the model is improved after adding the mixed attention mechanism.

#### 4.2.4. Visual Analysis of the Model

The feature information of interest to the network model can be seen from the visual feature map. In order to verify the attention of the added module ACmix to the small target feature, this paper visualizes the feature map output from the first stage and the last layer of the backbone network, as shown in Figure 12. It can be seen that in the first stage, the network focuses on the extraction of the overall features, while in the last convolution layer of the backbone network, it can be seen that the network focuses on some small target features.

#### 4.2.5. Receptive Field Improvement Experiment

The RFLA_Gauss_ module based on the Gaussian receptive field is added to the feature fusion area of the original network, and the performance of the original network is compared. The experimental results are shown in Figure 13 and Table 5. The GFLOPs of the model are reduced by 2.3% compared with the original network model, and other performances are not much different from the original model, indicating that the RFLA_Gauss_ module is added to the feature fusion area. The calculation performance of the model is improved.

#### 4.2.6. Overall Network Improvement

Based on the previous three groups of ablation experiments, the three models are combined into a completely improved network model in this paper. Compared with the performance of the original network model, the experimental results are shown in Figure 14 and Table 6. The average accuracy of the model class is 1.6% for AP, 2.9% for AP50, and 14.6% for AP50: 90, and the running speed of the algorithm is as high as 96 FPS. It basically meets the real-time requirements of urban traffic detection.

In this paper, Res3Unit is used to optimize the backbone network, and a hybrid attention mechanism is added to pay more attention to vehicle features and reduce the interference of other input features. The latest ACmix module is fused in the feature fusion area to enhance the receptive field of the network for small targets, and four groups of ablation experiments are carried out to prove the effectiveness of the improved model. Figure 15 shows the detection effect of the YOLOv7-RAR algorithm in different environments of urban roads. Table 7 shows that compared with other network models, our network model has a faster detection speed and higher detection accuracy.

## 5. Summary and Conclusions

In this paper, an accurate and real-time detection algorithm, YOLOv7-RAR, is proposed, and four groups of ablation experiments have been conducted successively. The experiments proved that YOLOv7-RAR could well realize vehicle detection with high accuracy and speed.

Through four groups of ablation experiments, this paper draws the following conclusions:Setting the structure of multiple fusion branches will reduce the use of repeated features in the network and fuse the features collected by the upper layer in a more fine-grained way.The separation of the attention mechanism module and the convolution module can extract the image features as much as possible, and the aggregation use can share the collected feature information to the greatest extent.Enhancing the receptive field for small targets in the distant view can reduce the miss rate of the model for vehicles in the distant view.By combining the three improved mechanisms, the final accuracy of the model is 2, which is 4% higher than that of the original model, and the AP50:90 performance is improved by 12.6% compared with the original algorithm.

The image data collected by the camera has a key impact on the prediction effect of the model. For places where the light is not good, the performance of the model will be poor. Solving the input problem of acquisition can make the application effects better.

We hope to continue to introduce faster and more accurate vehicle recognition algorithms in the future and to be able to use larger datasets and let the algorithm recognize more types of vehicles in order to hope to contribute to the field of vehicle detection. In addition, the transformer mechanism has ushered in another wave of research, which has potential research value in vehicle detection applications.

## Figures and Tables

**Figure 1 sensors-23-01801-f001:**
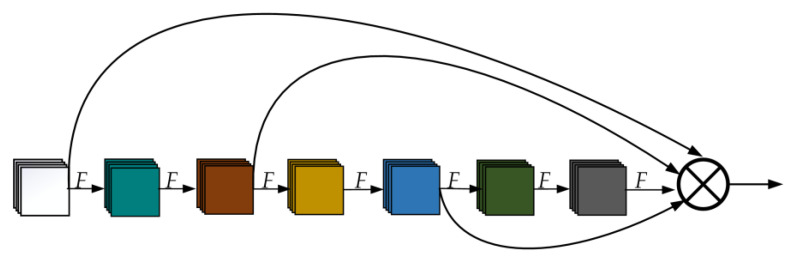
Schematic representation of the structure of the Res3Uint module, where *F* represents the convolution layer and **⨂** indicates concatenation.

**Figure 2 sensors-23-01801-f002:**
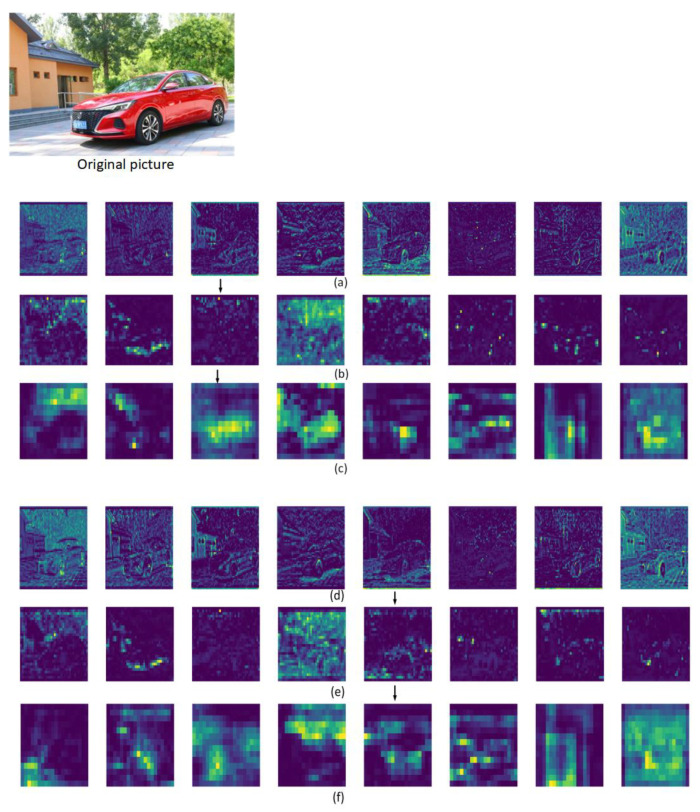
Fine-grained feature maps of the three stages in the backbone network that need to be sampled: (**a**) YOLOv7_algorithm_stage6_E-ELAN_features; (**b**) YOLOv7_algorithm_stage8_E-ELAN_features; (**c**) YOLOv7_algorithm_stage12_SPPCSPC_features; (**d**) our_algorithm_stage6_E-ELAN_features; (**e**) our_algorithm_stage8_E-ELAN_features; (**f**) our_algorithm_stage12_SPPCSPC_features.

**Figure 3 sensors-23-01801-f003:**
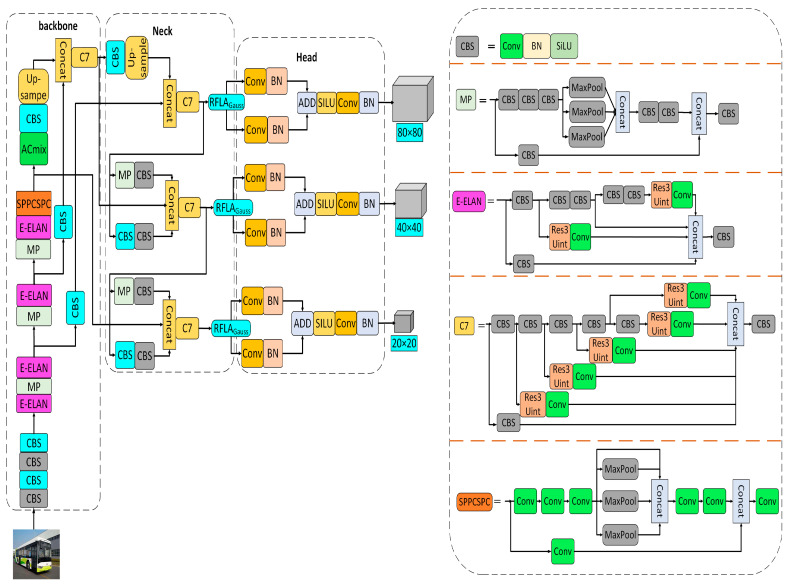
Model structure of the YOLOv7-RAR algorithm.

**Figure 4 sensors-23-01801-f004:**
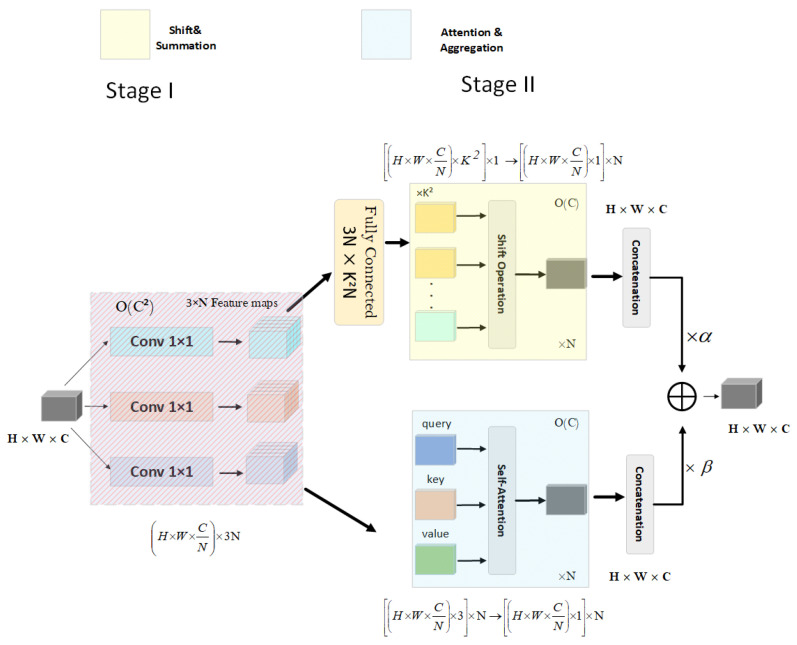
ACmix attention mechanism algorithm model.

**Figure 5 sensors-23-01801-f005:**
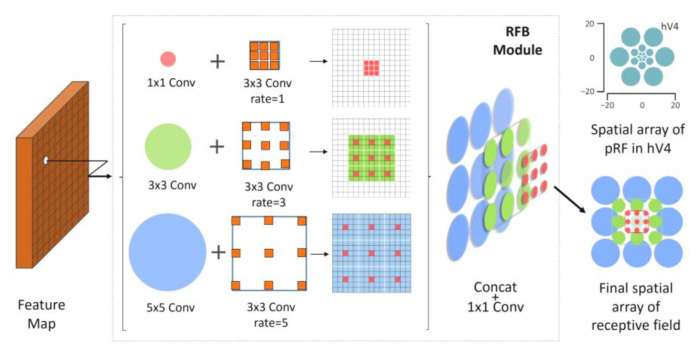
Mixed enhanced receptive field.

**Figure 6 sensors-23-01801-f006:**
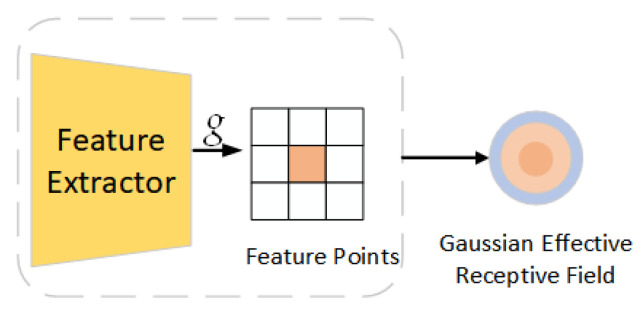
Feature schematic of the receptive field passing through the RFLA_Gauss_ module. *g* stands for the Gaussian convolution kernel.

**Figure 7 sensors-23-01801-f007:**
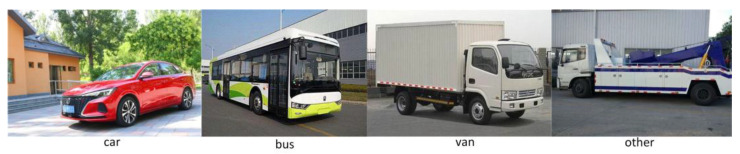
Types of images in the dataset, including cars, buses, vans, and other vehicles.

**Figure 8 sensors-23-01801-f008:**
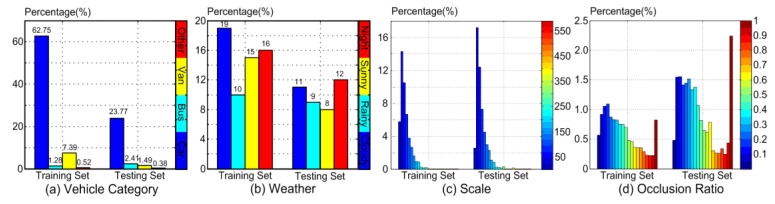
Shows the overall distribution of the dataset: (**a**) different vehicle types in the labeled dataset; (**b**) different environments in the dataset; (**c**) the scale distribution of the labeled dataset; (**d**) the occlusion ratio of different data.

**Figure 9 sensors-23-01801-f009:**
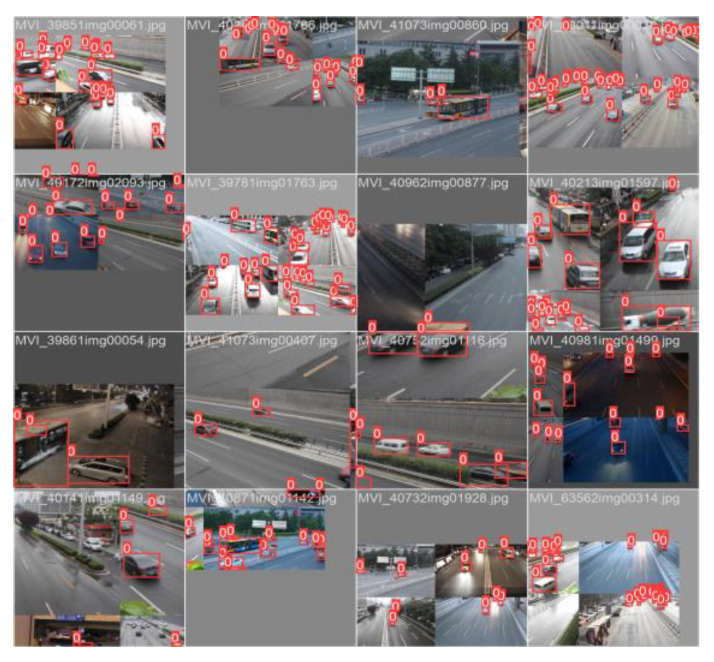
Target detection annotation samples for initial model training.

**Figure 10 sensors-23-01801-f010:**
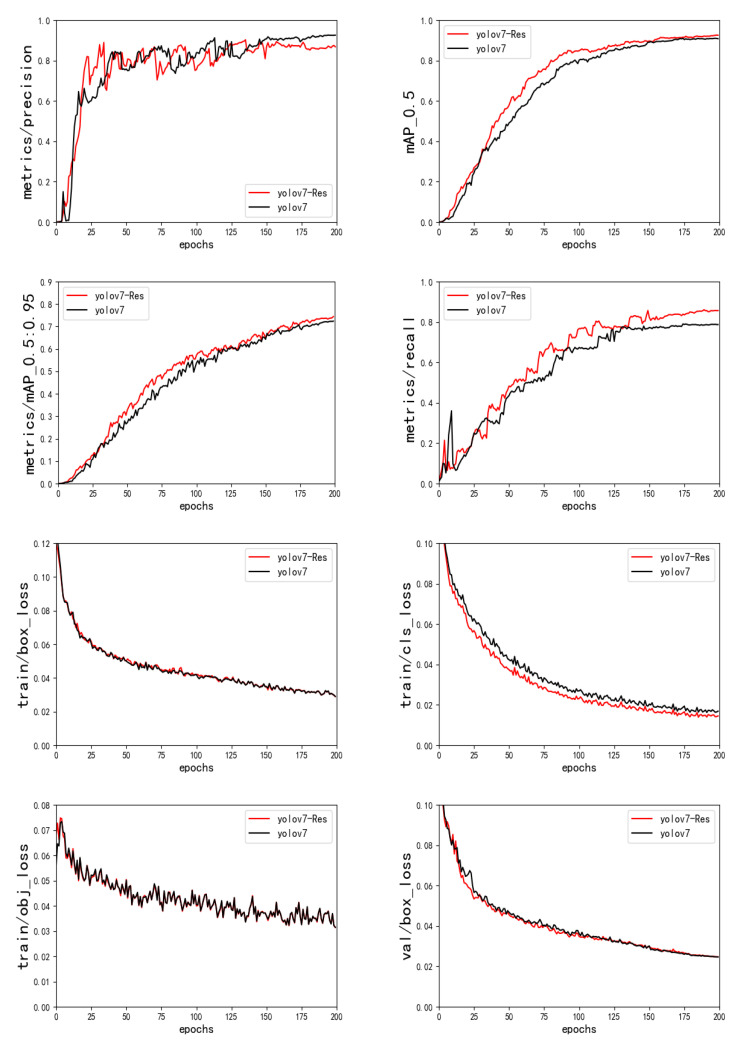
Results of the YOLOv7-Res3Unit ablation experiment.

**Figure 11 sensors-23-01801-f011:**
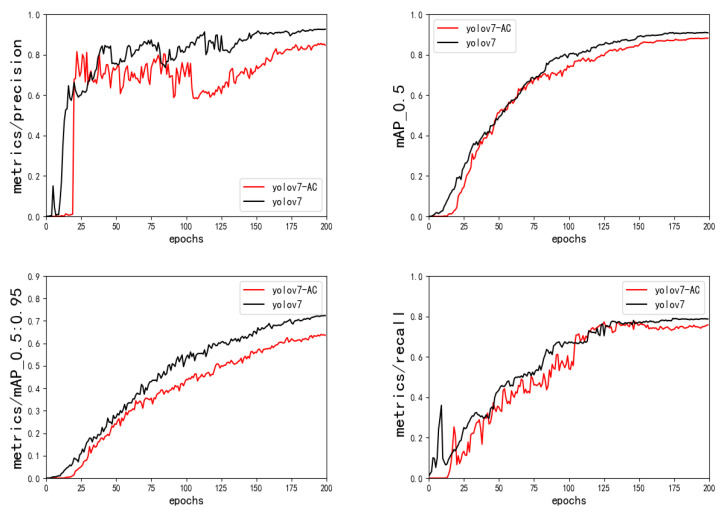
Results of the YOLOv7 ACmix ablation experiment.

**Figure 12 sensors-23-01801-f012:**
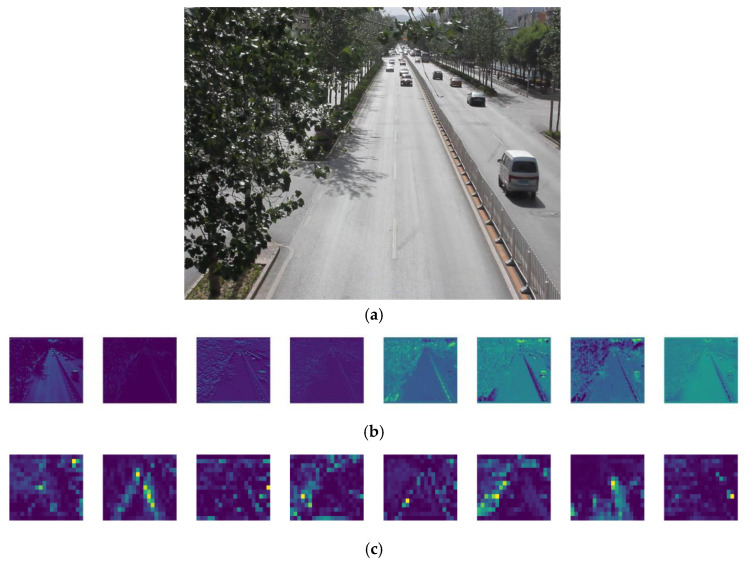
Visual feature map: (**a**) original image; (**b**) stage0_ Conv_ features; (**c**) Backbone_ Conv_ last layer_features.

**Figure 13 sensors-23-01801-f013:**
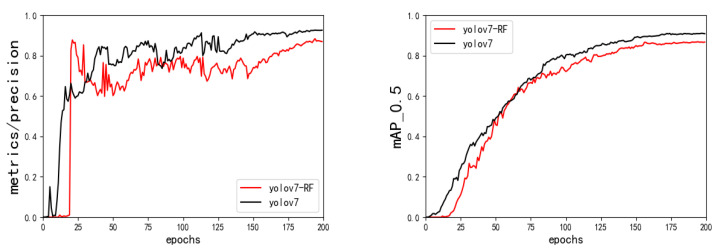
Results of the YOLOv7-RFLA ablation experiment.

**Figure 14 sensors-23-01801-f014:**
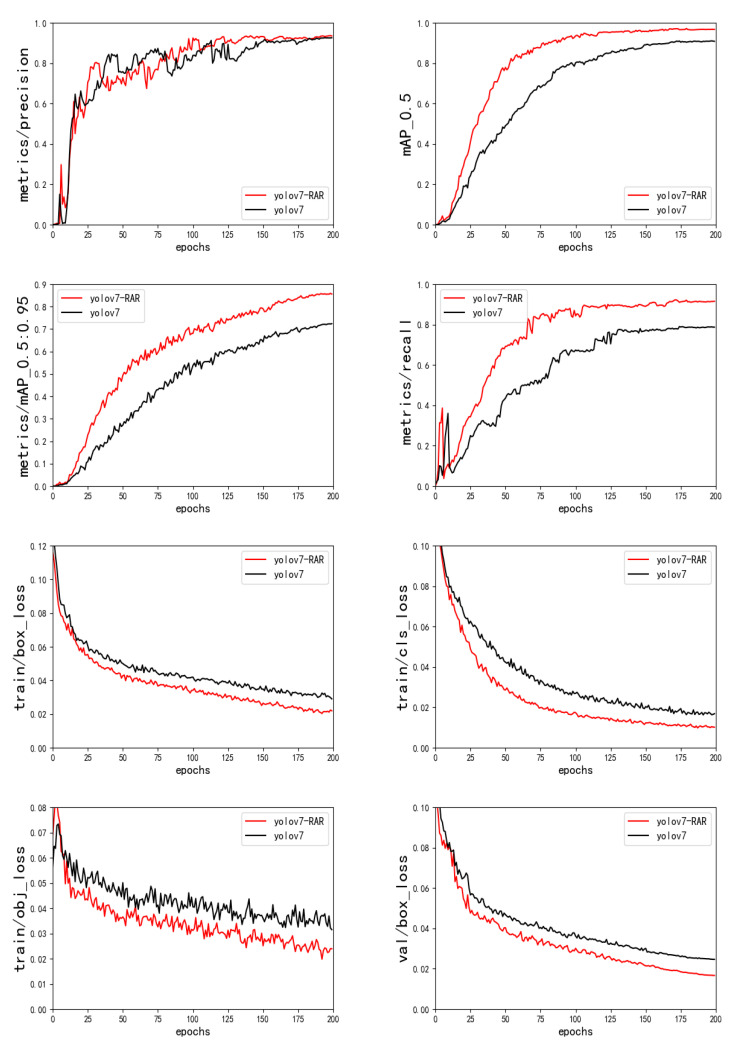
Results of YOLOv7-RAR ablation experiment.

**Figure 15 sensors-23-01801-f015:**
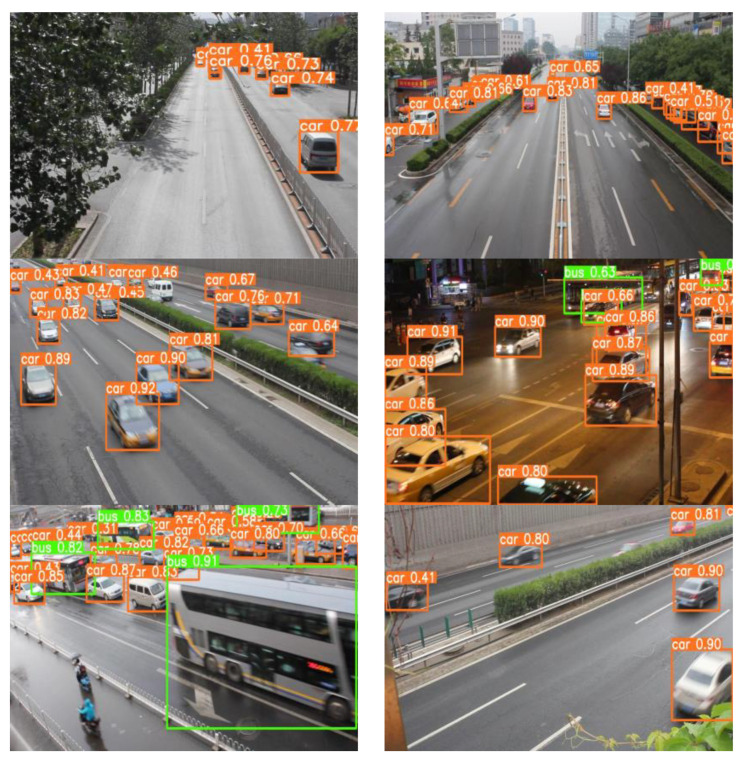
Prediction effect of the YOLOv7-RAR algorithm.

**Table 1 sensors-23-01801-t001:** Test performance of different algorithms on UA-DETRAC dataset.

Method	Input	mAP (%)	FPS	Environment
Faster R-CNN (VGG16)	-	72.70	11.23	GPU@A30
YOLO	448 × 448	62.52	42.34	GPU@M40
YOLOv2	416 × 416	73.82	64.65	GPU@A30
YOLOv2 544 × 544	544 × 544	75.96	39.14	GPU@A30
SSD300	300 × 300	74.18	58.78	GPU@A30
SSD512	512 × 512	76.83	27.75	GPU@A30
DSSD (ResNet-101)	321 × 321	76.03	8.36	GPU@TitanX
RefineDet320	300 × 300	76.97	46.83	GPU@A30
RefineDet512	512 × 512	77.68	29.45	GPU@TitanX
YOLOv3	416 × 416	88.09	51.26	GPU@TitanX
SpotNet	-	86.80	14	GPU@A30
YOLOv4	608 × 608	87.32	35	GPU@A30
YOLOv5s	640 × 640	92.46	55	GPU@A30
YOLOv7	640 × 640	92.70	151	GPU@RTX3090

**Table 2 sensors-23-01801-t002:** Initial parameters of the model.

Input Image Size	Initial Learning Rate	Batch Size	Momentum	Weight_Decay	Warmup_Epochs	Total Epoch
640 × 640	0.01	32	0.937	0.0005	3.0	200

**Table 3 sensors-23-01801-t003:** Network performance of YOLOv7-Res3Unit.

Model	AP/%	AP50/%	AP50:90/%	GFLOPs/G	FPS	#Param/M
YOLOv7	92.7	92.5	73.5	114.7	151	36.9
YOLOv7-Res	90.8	94.2	75.3	117.2	90	50.3

**Table 4 sensors-23-01801-t004:** YOLOv7-ACmix network performance.

Model	AP/%	AP50/%	AP50:90/%	GFLOPs/G	FPS	#Param/M
YOLOv7	92.7	92.5	73.5	114.7	151	36.9
YOLOv7-ACmix	86.3	91.3	66.1	102.2	122	34.2

**Table 5 sensors-23-01801-t005:** YOLOv7-RFLA network performance.

Model	AP/%	AP50/%	AP50:90/%	GFLOPs/G	FPS	#Param/M
YOLOv7	92.7	92.5	73.5	114.7	151	36.9
YOLOv7-RFLA	89.2	87.4	53.2	112.1	142	47.4

**Table 6 sensors-23-01801-t006:** YOLOv7-RAR network performance.

Model	AP/%	AP50/%	AP50:90/%	GFLOPs/G	FPS	#Param/M
YOLOv7	92.7	92.5	73.5	114.7	151	36.9
YOLOv7-RAR	95.1	98.3	86.1	135.1	96	65.4

**Table 7 sensors-23-01801-t007:** Some excellent algorithms are selected to be compared with our model.

Method	Input	mAP (%)	FPS	Environment
YOLO	448 × 448	62.52	42.34	GPU@M40
YOLOv2	416 × 416	73.82	64.65	GPU@A30
YOLOv2 544 × 544	544 × 544	75.96	39.14	GPU@A30
SSD300	300 × 300	74.18	58.78	GPU@A30
SSD512	512 × 512	76.83	27.75	GPU@A30
RefineDet320	300 × 300	76.97	46.83	GPU@A30
YOLOv3	416 × 416	88.09	51.26	GPU@TitanX
SpotNet	-	86.80	14	GPU@A30
YOLOv4	608 × 608	87.32	35	GPU@A30
YOLOv5s	640 × 640	92.46	55	GPU@A30
YOLOv7	640 × 640	92.70	151	GPU@RTX3090
YOLOv7-Res	640 × 640	90.80	90	GPU@RTX3090
YOLOv7-AC	640 × 640	86.30	122	GPU@RTX3090
YOLOv7-RF	640 × 640	89.20	142	GPU@RTX3090
YOLOv7-RAR	640 × 640	95.10	96	GPU@RTX3090

## Data Availability

Not applicable.

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
