# Peer review of "YOLOv7-RAR for Urban Vehicle Detection"

_sensors, 2023, doi:10.3390/s23041801_

Round 1

Reviewer 1 Report

The authors  presented a vechicle detection algorithm based on improved yolov7. This manuscript is not hard to follow. However, I have some minor concerns which need to be resolved.

1. In the abstract, the description of research objectives and methods should more closely connected and tighter.

2. In figure 2,  the layout of the diagram should be reasonable.

3. In figure 3, the words  are not clear.

4. The authors should again carefully check the whole paper in case of spelling mistakes.

Reviewer 2 Report

Please correct English and resubmit for review. The paper is hard to read and follow. 

i.e. line 67.  "we use CNN to extract features"  - Who is  we? This refer to work done by others? 

line 70. "The detection task is complete [12-16]." Not clear how this statement refers to cited papers. 

Reviewer 3 Report

1.      The paper proposed a Res3Unit structure is proposed to reconstruct the YOLOv7 backbone network, so as to improve the ability of network model architecture to obtain more nonlinear features. The aims and contributions in the abstract are confusing and (RAR and RFLA) abbreviations are not mentioned.

2.      Make a new section for related works after introduction. Describe all detailed techniques that have been used in urban vehicle detection by giving a detailed table of your survey including (dataset, models, and accuracy). Also, find a research gap and propose a solution for that.

3.      What do you mean by fine-grained features? Give a flowchart of improved Yolo-v7 using RAR and show the fine-grained features after layer modification in the backbone network.

4.      Explain the non linear feature extraction method.

5.      Why input image is subjected to four layers of convolution operations? Why not three or five layers?

6.      Figure 1 is not required. Instead, give a general block diagram of the proposed model.

7.      Did you try batch size other than 32? Did you try epoch’s number other than 200 rounds? Please justify your results.

8.      I think the input image size of 640*640 is too large to be trained by the model. Normally 224*224 is used with yolo networks.

9.      The learning rate is selected 0.01. Try 0.001 will give better detection rate.

10.  In the title you wrote Yolov7-RAR but in results section in figure 7, you wrote Yolov7-Res. Please unify them.  

11.  Compare your improved Yolov7-RAR to other related works that you have been surveyed in this paper.

12.  The first paragraph in Summary and Conclusions section is repeated in the abstract. Please remove it.

13.  Give some future directions at the end of section 4.

Reviewer 4 Report

Authors have used YoloV7 to detect urban vehicles.

Spacing between lines are more compared to original format.

Contributions should be added point wise.

Organization of paper can be added at the last of introduction.  

Authors have performed literature survey on object detection technique nicely. However, literature survey on vehicle detection needs to be added as it is the main work.

Resolution of figure 3 needs to improve.

In Section 3 title name, there is need of one space between number and title.

Authors have used only one dataset and section 3.1 is written datasets, it should be singular not plural.

It seems that authors have used only car as vehicle to detect in different weather conditions. If it is, title is not exactly matched. Authors have mentioned vehicle in title, it should be replaced with car. When vehicle is mentioned in title, reader understands different categories of vehicles. If it is not, mentioned all categories name with number of samples present in the dataset.

Results are nicely presented.

This manuscript needs to be formatted as per the journal requirements. 

Round 2

Reviewer 3 Report

no comments